# Bioactive Peptides and Proteins from Wasp Venoms

**DOI:** 10.3390/biom12040527

**Published:** 2022-03-30

**Authors:** Lei Luo, Peter Muiruri Kamau, Ren Lai

**Affiliations:** 1Key Laboratory of Animal Models and Human Disease Mechanisms of the Chinese Academy of Sciences/Key Laboratory of Bioactive Peptides of Yunnan Province, KIZ-CUHK Joint Laboratory of Bioresources and Molecular Research in Common Diseases, National Resource Center for Non-Human Primates, Kunming Primate Research Center, National Research Facility for Phenotypic & Genetic Analysis of Model Animals (Primate Facility), Sino-African Joint Research Center, and Engineering Laboratory of Peptides, Kunming Institute of Zoology, Kunming 650107, China; luolei@mail.kiz.ac.cn (L.L.); peter@mail.kiz.ac.cn (P.M.K.); 2University of Chinese Academy of Sciences, Beijing 100049, China; 3Southern Marine Science and Engineering Guangdong Laboratory (Guangzhou), Guangzhou 511458, China

**Keywords:** wasp, venom, peptides, proteins, pharmaceutically active molecules

## Abstract

Wasps, members of the order Hymenoptera, use their venom for predation and defense. Accordingly, their venoms contain various constituents acting on the circulatory, immune and nervous systems. Wasp venom possesses many allergens, enzymes, bioactive peptides, amino acids, biogenic amines, and volatile matters. In particular, some peptides show potent antimicrobial, anti-inflammatory, antitumor, and anticoagulant activity. Additionally, proteinous components from wasp venoms can cause tissue damage or allergic reactions in organisms. These bioactive peptides and proteins involved in wasp predation and defense may be potential sources of lead pharmaceutically active molecules. In this review, we focus on the advances in bioactive peptides and protein from the venom of wasps and their biological effects, as well as the allergic reactions and immunotherapy induced by the wasp venom.

## 1. Introduction

Wasps belong to the order Hymenoptera, the third-largest order of insects. The wasps comprise an incredibly diverse group in the suborder Apacrita, classified into two classes; the Parasitica and Aculeata [1]. The clade Aculeata is further divided into solitary and social wasps according to their social lifestyle [2]. The social wasps generally use their venom for defense and self-preservation [3]. The social wasp’s venom has evolved over countless years to make its bites more painful and its immune and allergic responses more intense than those of solitary wasps [3]. The chemical constituents of the venom of social wasps are well documented, including allergens, enzymes, bioactive peptides, and amine substances [4,5]. Solitary wasps tend to use their venom for predation rather than defense. Most solitary wasps use venom-containing neurotoxic and antibacterial components to quickly paralyze their prey, and then save the prey for larvae to eat.

The prevention and treatment effects of wasp venom on the rhinitis, rhinoconjunctivitis, rheumatoid arthritis, ischemia stroke, Alzheimer’s disease, Parkinson’s disease, and epilepsy have been gradually improving [6,7,8,9,10,11]. Other potential therapeutic activities such as anticancer activities are also being investigated [12]. Therefore, wasp venom is an essential reservoir of pharmacologically active molecules [13]. However, to date, some of the constituents of wasp venom remain unexplored. This review summarizes the representative bioactive peptides and proteins identified from vespid venom.

## 2. General Origins and Properties of Wasp Venoms

Venomous animals secrete venom to rapidly immobilize and inactivate their prey or predator. The venom constituents target critical systems, such as the cardiovascular, nervous, and immunological systems, to achieve effective and rapid immobilization or defense. Considering that venomous animals prey on various species and have defensive systems against predators, they produce multiple peptidic and proteinous molecules with distinct molecular targets [14]. The significant similarities demonstrated in the protein constituents between the venoms from different animals clearly indicate that throughout evolution, other protein molecules have been regularly and convergently recruited in venoms, whereby those appropriate for recruitment have been under functional and/or structural constraints. [15,16].

Wasp venoms, particularly those of the well-studied social Vespidae, often induce local reactions such as edema, pain sensation, and wheal. These reactions may be mediated by various bioactive molecules, including chemotactic peptides, mastoparans, and bradykinin-like peptides. [17,18,19]. Besides the direct effects of wasp stings, immunologic responses that typically result in anaphylaxis and severe anaphylactic shock, and the allergens, such as hyaluronidase, cysteine-rich secretory proteins, serine proteases, phospholipase, pathogenesis-related proteins, and Antigen 5, have been studied extensively. Systemic toxic effects such as acute renal failure, rhabdomyolysis, hemolysis, coagulopathy, aortic thrombosis, and other effects can occur in response to extensive envenomation by a wasp. [20].

### 2.1. Wasp Venoms Components

Wasp venoms comprise a variety of biomolecules with therapeutic potential that play a part in crucial activities necessary for the survival of the wasp. These biomolecules include (a) proteins such as enzymes and allergens; (b) short peptides with different functions such as antibacterial and neuroprotective properties; and (c) small molecules with low molecular weight such as bioactive amines, amino acids, and so on [13] (Figure 1).

### 2.2. Peptidic Components

The majority of the peptide components of the venom include amphipathic peptides with cationic groups and helical structures, ranging in length from 12 to 50 amino acids. Fifty percent of these components interact with cell membranes using their hydrophobic residues [21], accounting for seventy percent of the wasp venom’s dry weight [21,22]. Well-known peptidic components from wasps’ venoms include neurotoxic peptides, kinins, mastoparans, and chemotactic peptides.

#### 2.2.1. Neurotoxic Peptides

In wasp venom, many neurotoxic peptides that modulate the activity of various receptors, including ion channels, have been discovered and characterized (Table 1). The first neurotoxic component identified from wasp venom was a kinin that irreversibly blocks the nicotinic synaptic transmission by depleting the transmitter at the presynaptic site [23,24,25]. Progressively, Tom Piek et al. reported a Threonine^6^-bradykinin in the venom of the wasp *Colpa interrupta* (F.), which presynaptically blocked nicotinic synaptic transmission in the Central Nervous system (CNS) of insects [26]. It has been gradually established that many kinins contained in Vespidae wasp venom are responsible for paralysis [4,27]. To date, most of the neurotoxic peptides of hunting wasps are kinins.

Two neurotoxic non-kinins, α-PMTX and β-PMTX, were identified and purified from *Anoplius samariensis* and *Batozonellus maculifrons*, respectively [28,29]. PMTXs, which are 13-amino-acid venom peptides, target the neural systems of both vertebrates and invertebrates by inhibiting or slowing the inactivation of sodium channels [30]. α-PMTX significantly increases neurotransmission in the lobster leg neuromuscular junction [29]. Notably, the efficacy of β-PMTX increased by five times compared to that of the parent peptide, α-PMTX. β-PMTX is formed by substituting the lysine at position 12 with an arginine [28]. In addition, two new neurotoxin’s peptidic molecules, known as AvTx-7 and AvTx-8, have also recently been identified from *Agelaia vicina* [31,32]. AvTx-7 promotes glutamate release via the K^+^ channel, while AvTx-8 blocks GABAergic neurotransmission.


biomolecules-12-00527-t001_Table 1Table 1Representative Neuropeptides from Wasp-venom.Wasp-Scientific NameIsolated PeptidesAA SequenceReference
*Sphex argentatus*
Sa12bEDVDHVFLRF[33]
*Anoplius samariensis, Batozonellus maculifrons*
α-PMTXsRIKIGLFDQLSKL[28]
*Anoplius samariensis, Batozonellus maculifrons*
β-PMTXsRIKIGLFDQRSKL[28]
*Batozonellus maculifrons*
β-PMTXsRIKIGLFDQLSRL[5]
*Agelaia vicina*
Agelaiatoxin-8 (AVTx8)INWKLGKALNALL[34]


#### 2.2.2. Kinins

Bradykinin, a mammalian blood serum molecule, was discovered in 1949, and experimentally, it was reported to cause the ileum contraction of guinea pigs. [35]. This molecule generally induces contractions or relaxations in smooth muscles. By depolarizing nerve terminals in vertebrate neurons, bradykinin causes the release of neuropeptides, galanin, vasoactive intestinal peptide, and neuropeptide Y, as well as catecholamines dopamine, adrenaline, and norepinephrine [36,37].

The first component discovered in wasp venom was a kinin-like or BK-related peptide (BRP), characterized as a pain- inducing molecule [38]. Since then, many peptides from this class have been discovered in social wasp venoms and are generally recognized as wasp kinins. Although there is a notable similarity in kinins sequences from wasp and that of the mammalian bradykinin [-PPGF (T/S) P(F/L)-], several kinins have lengthier sequences or diverge at positions 3 and 6, where the proline residue is substituted by hydroxyproline and the serine residue by threonine (Table 2). Thr^6^-bradykinin exerted a 3-fold more substantial anti-nociceptive effect in rats and lasted longer than bradykinin following substitution of single amino acid [39]. Bradykinins from solitary wasp venoms probably paralyze prey during hunting through depletion of neurotransmitters of the nicotinic acetylcholine receptor in the insect’s central nervous system [24,25,27]. Almost all social wasp venom undoubtedly contains kinin-like activity; however, only *Cyphononyx dorsalis* (Pompilidae) and a few Scoliidae wasps have been reported to contain Thr^6^-bradykinin in their venom, while bradykinin was only found in *Megacampsomeris prismatica*’s venom (Scoliidae) [27,40]. Additionally, vespakinin M (amino acid sequence GRPPGFSPFRID) is the first kininogen isolated from invertebrates from *Vespa magnifica* venom. Interestingly, vespakinin M is the first kininogen identified from insects and invertebrates [41].

#### 2.2.3. Mastoparans

The mastoparans are the most abundant peptide components of the hunting wasp venom and are widely distributed in Vespidae: social and solitary (Table 3). Commonly, mastoparans are tetradecapeptides, and they are known to act on mast cells resulting in an inflammatory response [47]. Additionally, mastoparans have structural features, including an amphiphilic α-helix structure and a net positive charge, in which all hydrophobic amino acid side chains are on one side and those of basic or hydrophilic amino acid residues on the opposite side. This enables the mastoparans to attach to the biomembranes and form pores, thus increasing cell membrane permeability [48].

Mastoparans bind to the membranes of fungi, bacteria, erythrocytes, and mast cells, exhibiting hemolytic, antibacterial, and mast cell degranulating (MCD) activity. MCD activity, as caused by mastoparan, also may occur through granule exocytosis, which is initiated by mastoparan modulating G protein function without receptor association [49]. The overall effect of MCD activity is cell type-dependent: histamine is released by mast cells, platelets release serotonin, pancreatic β-cells release insulin, and catecholamines are released by chromaffin cells [47,50,51]. Additionally, the mastoparans’ lysis of cells varies according to cell type. In general, mastoparans have a larger antimicrobial property against fungi than Gram-negative bacteria such as *E. coli* [52,53]. Moreover, mastoparans have been found to cause feeding disorder in lepidopteran larvae, most likely due to their non-specific neurotoxic or myotoxic action-induced lytic action against insect cells [52].

Additionally, cell lysis of mastoparan, results in an increase in tumor cell cytotoxicity, a mitochondrial permeability change, and the impairment of cell viability [12]. With MCD and cell lysis activity, mastoparans induce phospholipases A, C, and D, and mobilize calcium from the mitochondria and sarcoplasmic reticulum, causing necrosis and/or apoptosis [12,54]. Mastoparans’ diverse biological roles have sparked interest in their possible therapeutic and biomedical applications [12,55]. Nevertheless, they have not been put to use due to a lack of specificity, as not onlywere they reported to damage cancer cells, but they also show a harmful effect on healthy cells. Thus, efforts should be made to develop a delivery method for venom peptides that specifically targets cancer [56].


biomolecules-12-00527-t003_Table 3Table 3Representative Mastoparans from Wasp-venom.Wasp-Scientific NameIsolated PeptidesAA SequenceReferences
*Vespa xanthoptera*

*Vespula lewisii*
Mastoparan INWKGIAAMAKKLL[46,57]
*Vespula vulgaris*
Mastoparan V1INWKKIKSIIKAAMN[57,58,59]
*Vespa magnifica*
Peptide 12aINWKGIAAMAKKLL[53]
*Vespa magnifica*
Peptide 12bINWKGIAAMKKLL[53]
*Vespa magnifica*
Peptide 12dINLKAIAAMAKKLL[60]
*Vespa basalis*
Mastoparan BINLKAIAAFAKKLL[61]
*Vespa tropica*
Mastoparan-VT1INLKAIAALAKKLL[62]
*Vespa basalis*
Mastoparan-AIKWKAILDAVKKVI[61]
*Vespa affinis*
Mastoparan-AFINLKAIAALAKKLF[63]
*Vespa basalis*
Mastoparan-BLKLKSIVSWAKKVL[64]
*Vespa bicolor*
Mastoparan-VB1INMKASAAVAKKLL[65]
*Vespa crabro*
Mastoparan-CLNLKALLAVAKKIL[66]
*Vespa ducalis*
Mastoparan-DINLKAIAAFAKKLL[63]
*Vespa velutina*
Mastoparan-VIAWKGIAAMAKKLL[63]
*Vespa xanthoptera*
Mastoparan-XINWKGIAAMAKKLL[67]
*Polybia paulista*
Polybia-MP IIDWKKLLDAAKQIL[46]


#### 2.2.4. Chemotactic Peptides

Chemotactic peptides come second as the major category of peptidic molecules found in wasp hunting venom (Table 4). As with mastoparans, chemotactic peptides are exclusively found in social and solitary Vespidae wasp venoms. On the other hand, chemotactic peptides are generally tridecapeptides having an amphipathic, α-helical, linear, cationic, and C-terminal amidated structure. They elicit a chemotactic reaction, particularly in polymorphonuclear leukocytes and macrophages [68]. Given their structural similarity, chemotactic peptides frequently exhibit mastoparan-like MCD, hemolytic, and antibacterial effects. A minor edematogenic response occurs in response to the chemotactic activity, which is commonly followed by an inflammatory exudate around the envenomation site, with a high concentration of polymorphonuclear leukocytes. Because chemotactic peptides enhance the inflammatory response caused by wasp stings rather than directly triggering nociception [13], they are presumably implicated in defense, and this hypothesis is supported by their broad distribution in the venoms of most social wasps.

Overall, no conserved primary structure for detecting venom-chemotactic peptides has been identified to date. Conversely, the majority of sequences indicate representative motives, ZZ(G/R)ZZ, ZZ(G/A/S/R/K/T)(G/T/K/S)ZZ, or sometimes overlapped version of the two motives, at which Z is a hydrophobic amino acid, most typically Isoleucine or Leucine. Chemotactic peptides, unlike mastoparans, have these motives. In addition, chemotactic peptides, usually have only one or two Lysine residues, whereas most mastoparans have two or three residues [69].

Ombati et al. identified VESCP-M2, a membrane disrupting toxin, from the venom of *Vespa mandarinia* [70]. In comparison to analogues of similar sequences, VESCP-M2 contains a positively-charged α-helix structure. VESCP-M2 has been implicated in causing tissue injury symptoms such as instantaneous pain, pruritus, and dermal necrosis in response to *Vespa mandarinia* envenomation. In addition, electrophysiological assays demonstrated that VESCP-M2 has a strong ability to permeate the biological membranes [70]. Also, Xueqing Xu et al. report two families of antimicrobial peptides from *Vespa magnifica* (Smith). The primary structures of Peptide5e, Peptide 5f, and Peptide 5g are homologous to those of chemotactic peptides [53]. Orancis-Protonectin, a bioactive peptide with increased hemolytic activity, was identified from *Orancistrocerus drewseni* Eumenine venom. The sequence is remarkably similar to protonectins, hemolytic peptides from social wasp venoms [71].


biomolecules-12-00527-t004_Table 4Table 4Representative Chemotactic Peptides from Wasp-venom.Wasp-Scientific NameIsolated PeptidesAA SequenceReference
*Vespa mandarinia*
VESCP-M2FLPILAKILGGLL[70]
*Vespa magnifica*
VCP-5hFLPIIGKLLSGLL[72]
*Vespa bicolor*
VESP-VB1FMPIIGRLMSGSL[65]
*Vespa xanthoptera*
VesCP-XFLPIIAKLLGGLL[73]
*Vespa magnifica*
Peptide 5eFLPIIAKLLGGLL[53]
*Vespa magnifica*
Peptide 5f FLPIPRPILLGLL[53]
*Vespa magnifica*
Peptide 5g FLIIRRPIVLGLL[53]
*Vespa magnifica*
Peptide 5hFLPIIGKLLSGLL[53]
*Vespa analis*
VesCP-AFLPMIAKLLGGLL[73]
*Vespa mandarinia*
VesCP-M FLPIIGKLLSGLL[73]
*Vespa orientalis*
HR-IIFLPLILGKLVKGLL[73]
*Vespa tropica*
VesCP-TFLPILGKILGGLL[73]
*Vespa tropica*
VCP-VT1FLPIIGKLLSGLL[62]
*Vespa tropica*
VCP-VT2FLPIIGKLLSG[62]
*Vespa crabro*
CrabrolinFLPLILRKIVTAL[74]
*Polybia paulista*
Polybia-CPILGTILGLLKSL[46]
*Orancistrocerus drewseni*
Orancis-ProtonectinILGIITSLLKSL[71]


#### 2.2.5. Other Peptides

More peptides from wasp venoms have been previously reported. Notably, Vespin, a bioactive peptide, was identified from the venom of *Vespa magnifica* (Smith). Its amino acid sequence contains 44 residues, with 15 being leucines or isoleucines (32%). Vespin demonstrated a contractile activity on isolated ileum smooth muscle. [75].

### 2.3. Proteins Components

Most of the proteins defined by wasp venoms have been shown to be lethal and hasten fatal allergic reactions, including anaphylaxis and tissue damage [21,22]. The most common allergens found in wasp venoms include antigen 5, phospholipase A1, and hyaluronidase, which have been detected from wasp venoms of all species (Table 5).

#### 2.3.1. Hyaluronidase

Hyaluronidase, a 33–45 kDa glycoprotein, is a primary allergen of wasp venom and is vital for the cross-reactivity of hymenopterans (wasp and bee) venoms with allergic patient sera [18]. The vespid hyaluronidase is a member of family 56 of glycosyl hydrolase with endo-N-acetylhexosaminidase enzymatic specificity [96], and is known to stimulate the secretion of reducing groups from hyaluronic acid [97]. Thus, the venom’s hyaluronidase enzyme leads to the degradation of hyaluronic acid following the wasp’s sting. Numerous investigations have established a significant degree of sequence similarity between the various hyaluronidases found in bee or wasp venoms [98,99]. Additionally, hyaluronidase has been characterized as an allergenic factor in multiple wasp species that elicits fatal anaphylactic reactions via IgE in human subjects. The high protein concentration and its activity are thought to contribute to the venom’s allergic and poisonous potential [78]. For instance, *V. tropica* contains 2.5 times the amount of hyaluronidase reported in *V. affinis*. In turn, hyaluronidase, phospholipase, and dipeptidyl peptidase activity were increased, with the phospholipase, and dipeptidyl peptidase having equal proportions, implying that *V. tropica* venom is more potent in inducing allergic reactions and poisoning than *V. affinis* venom.

#### 2.3.2. Phospholipases

The phospholipases found in wasp venoms have been extensively described [100]. These proteins catalyze the hydrolysis of membrane phospholipids containing diacylphospholipids, like phosphatidylserine, phosphatidylcholine, and phosphatidylethanolamine, resulting in the formation of lysophospholipids and free chain fatty acids. Due to the vast array of symptoms mediated by phospholipases, they play an essential role in the insect sting-triggered envenoming process, especially on Hymenoptera venom allergy (HVA). In addition, these molecules have a variety of direct pathophysiological effects, such as necrosis, cell membrane disruption, inflammation, hemolysis, pore formation, platelet aggregation activation (PLA1), and possibly apoptosis (PLA2) [101].

It is hypothesized that phospholipase can destabilize the phospholipid packing of a wide variety of cellular membranes, resulting in severe hemolysis, heart failure, and fatality in both animals and humans. PLA1, a non-glycosylated protein with a molecular mass of 34 kDa, acts as a toxin by hydrolyzing sn-1 fatty acids in phospholipids, thereby generating free fatty acids (e.g., arachidonic acid) and 2-acyllysophospholipid.

The PLA1s represent 6–14 percent of the vespid venom dry mass and have been previously identified in several northern hemisphere wasp species and ants. Lately, a PLA1 with 304 amino acids was isolated and described from the venom of *Polybia paulista*. Numerous PLA1s from wasp venom have been reported as essential allergens and important factors in inflammatory processes [83,102,103]. Recently, magnifin, a PLA1 was identified from the venom of *V. magnifica*. Magnifin robustly caused platelet aggregation and thrombosis in vivo at low doses [83].

#### 2.3.3. Antigen 5

Along with PLA1 and Hyaluronidase, antigen 5 is a significant venom allergen found in nearly all allergy-causing Vespoidea species. Antigen 5, a member of a superfamily of secreted proteins, is a non-glycosylated protein with a molecular weight of 23 kDa. It belongs to the superfamily of catabolite activated proteins (CAP), which is composed of cysteine-rich secretory proteins (CRISP) [21]. This protein has been identified as a powerful allergen and frequently the most potent of all allergens in patients following the sting of these Vespidae members. Numerous antigen 5 variants have been identified, and the allergenic activity and sequence similarity have been extensively studied [88,104,105]. Most vespid-allergic patients exhibit numerous reactions to multiple vespid venoms [106,107], indicating that the component proteins are partially antigenic. Bees, fire ants, and vespids all contain venom allergens that are distinct from one another. Two of the four allergens found in fire ants are homologues to vespid phospholipases and antigen 5. Antigen 5 from fire ants has an approximate 35% sequences identity match with antigen 5 from vespid [108]. Antigen 5 from vespids shares partial sequence identity in their C-terminal region with proteins identified from numerous sources, including nematode, glioma, mouse and human testis, tomato, and lizard [109,110,111]. The phospholipases or Antigen 5 of yellow jacket wasps and hornets have shown a 44–68 percent sequence identity, whereas their hyaluronidases have a 73–92 percent sequence identity [108]. Antigen 5 sequence similarity for species within the same genus is 98 percent but lowers to 57 percent when antigen 5s from different genera, for instance, Vespula and Polistes, are compared [88]. Each of these antigen 5s exhibits some degree of immunological cross-reactivity, which is important in defining the composition of mixtures for desensitization treatment [88,112].

#### 2.3.4. Other Protein Components

As members of the S1 trypsin family of the SA clan, serine proteases are the most prominent family of peptidases [18]. These proteins affect the homeostatic system, impacting various coagulation molecules, the fibrinolytic system, and individual cells. Additionally, some of these proteases have been shown to cause allergic reactions or hinder melanization, serving as a neurotoxin. Junyou Han and colleagues isolated and characterized Magnvesin, an anticoagulant serine protease, from *Vespa magnifica* wasp venom. Magnvesin is a protein that inhibits blood coagulation and has a serine protease-like function. Magnvesin is 52 percent identical to the serine protease from *Polistes dominulus* [93].

Rungsa et al. also identified two proteins from *V. tropica* and *V. analis* that matched *V. basalis* dipeptidyl peptidase IV (DPPIV). Additionally, DPP-IV is a serine protease that causes degradation to dipeptides from the N-terminal, including proline or alanine. Even though the function of DPP-IV is obscure, it is associated with exosomes and could be limited to the activating or inactivating venom molecules, which may produce additional or synergistic effects to accelerate the venom’s toxicity [91].

Three proteins have been discovered and reported from the solitary spider wasp *Cyphononyx dorsalis*, a well-known spider predator. These proteins included an elastase-like protein with a high degree of homology to the fire ant, an arginine kinase-like protein with a high degree of homology to the honeybee, and an unknown protein with no homologs in the database. By using the purified proteins, the bioassay results suggested that the arginine kinase-like protein was responsible for spider wasp paralysis [113].

Furthermore, Zhe Lin et al. reported 75 potential protein molecules from *M. mediator* identified through a comprehensive transcriptome and proteomic research approach. The identified venom components, such as glycoside hydrolase family 18 enzymes, metalloproteases and serine protease inhibitors, were consistent with previously described proteins from other parasitoid wasps. Further investigation revealed that 511 differentially expressed proteins (DEP) are predominantly involved in extracellular matrix receptor interaction, immunological response, and material metabolism [114].

## 3. Pharmacological and Medical Application of Bioactive Peptides from Wasp Venom

Given the complexity and diversity of the components of the wasp venom and its preventive and therapeutic effects on a variety of diseases, wasp venom can serve as a candidate resource pool for novel therapeutic molecules [13]. Although some candidate molecules with excellent therapeutic effects have been discovered, further exploration of Vespa toxins is still needed. Here, we summarise some bioactive peptides from wasp venom, including antimicrobial, antitumor, anti-inflammatory, and anticoagulant peptides (Figure 2).

### 3.1. Antimicrobial Activities

Antimicrobial peptides (AMPs) are a kind of basic polypeptide substance with relatively small molecular weight (<10 kDa), which are generally composed of 20–60 amino acid residues and have little or no capacity to induce antimicrobial resistance [115]. Most of these active polypeptides have the characteristics of strong basic, thermal stability and broad-spectrum antibacterial effect [116,117,118]. The AMPs show a high-efficiency broad-spectrum killing effect on both gram-negative and gram-positive bacteria. Some AMPs also display antimicrobial activity against viruses, fungi, and protozoa [119,120]. Several AMPs, which mainly belong to the classes of mastoparans, kinins, and venom chemotactic peptides, have been identified from many vespid wasp species [62].

Wasp-derived mastoparan and mastoparan-like peptides showed potent antibacterial activity. However, none exhibited good selectivity to bacterial and mammalian cell membranes [58,121,122]. Several studies on mastoparan-related peptides have shown that the mastoparan has strong antibacterial activity and causes more significant damage to the cell membrane. For instance, the minimal inhibitory concentration (MIC) values of mastoparan-X were 2.5 μM for *Lactococcus lactis* and 8 μM for *E. coli*, respectively [123]. After side-chain hydrophobicity modulation, the mastoparan-X analogues, Adec1, Adec8, and Adec14 exhibited more potent antimicrobial activities than the parent structure. For Adec1, the MIC values were 1.7 μM for *Lactococcus lactis* and 3.3 μM for *E. coli*, respectively. However, the H5 values, at which 5% haemolysis has been obtained, were 18 μM for mastoparan-X and 0.7 μM for Adec1, indicating that the stronger the antibacterial activity, the bigger the damage to the cell membrane [123]. Therefore, structural modifications and functional optimization of mastoparan peptides are essential to reduce their side effects and enhance their potential for clinical application [12]. Antibacterial peptides with little hemolytic activity have also been purified and characterized from the wasps’ venom [12,53]. Unlike the mastoparan peptides, the antibacterial activity of chemotactic peptides has been rarely reported. In our previous study, three chemotactic peptides were purified and characterized from *V. magnifica* venom, exhibiting antibacterial activity against bacteria and fungi. For peptide 5e, the MIC values for *E. coli*, *Staphylococcus aureus,* and *Candida albicans* were 30, 5, 25 μg/mL, respectively. For peptide 5f and peptide 5g, MIC values against *E. coli*, *S. aureus* and *C. albicans* were similar to peptide 5e. All three wasp chemotactic peptides show typical antibacterial peptide characteristics: including residues rich in positive charges and amphiphilic structures [53].

### 3.2. Antitumor Activities

Mastoparan is a class of amphiphilic cationic polypeptides that can also be used as antitumor agents [124]. After being specifically transferred into the tumor cells, the mastoparan could induce mitochondrial permeability transition to kill cancer cells rather than normal cells. There are also mastoparans from other wasps showing antitumor activities. Both *V. crabro* and *V. analis* mastoparans exhibit anticancer effects on ovarian tumor cells. 100 μM of *V. crabro* mastoparan caused a dramatic decrease (almost 80%) in the relative survival fold of SK-OV-3 cells, while 100 μM of *V. analis* mastoparan treatment induced an extremely low survival fold (30%) of SK-OV-3 cells [125]. MP1, another mastoparan-related peptide, was reported to specifically kill cancer cells such as prostate cancer cell line PC-3 (cell proliferation MTT assay IC_50_ = 64.68 μM) and bladder tumor cells Biu87 (IC_50_ = 52.16 μM) and EJ (IC_50_ = 75.51 μM) [126] as well as multidrug-resistant leukemic cells K562/ADM (IC_50_ = 26.55 μM) [127]. Altogether, wasp-derived mastoparan-related peptides may become candidate lead molecules to develop novel anticancer drugs [128].

Although wasp-derived mastoparan-related peptides have shown potent therapeutic effects on cancer and are expected to become alternative therapies for cancer treatment, their application still faces several obstacles. For instance, the instability toward proteases and anticancer activity of MP1 needs to be further improved [129]. Therefore, chemical modifications and substitutions are required for better pharmacological properties of antitumor peptides. After being substituted with thioamide bond, MPI-1 demonstrated more robust anticancer activity (for PC-3, IC_50_ = 20.3 μM, for EJ, IC_50_ = 21.6 μM) and weaker side effects in vitro and in vivo [130]. The synthetic variants of decoralin, a natural AMPs from wasp *Oreumenes decorates*, exhibits potent antitumor activities against MCF-7 breast cancer cells (IC_50_ = 12.5 μM) [131]. Both the conformational constraint and targeted delivery system improve the therapeutic effect and reduce cytotoxicity [132,133,134], which further reinforces the importance of conformational modification and drug delivery systems.

### 3.3. Anti-Inflammatory Activities

Inflammation is a defensive response of the body to stimuli and is associated with many diseases such as arthritis, rheumatoid arthritis, and asthma. Existing anti-inflammatory drugs inevitably have side effects on the gastrointestinal tract, kidney, central nervous system, and blood system. Therefore, it is essential to tap new molecules with better anti-inflammatory properties and fewer side effects. Both the in vivo and in vitro experiments demonstrate that wasp venom could significantly ameliorate inflammatory response symptoms and release of inflammatory factors. For example, Masroparan-1, a tetradecapeptide isolated from wasp, significantly suppresses TNF-α, interleukin-6 (IL-6), and IL-1β. 40 μM Masroparan-1 almost completely reduced the mRNA expression of TNF-α and IL-6 [135]. Although crude wasp venom contains components that elicit toxic reactions, the crude venom also exhibits potent anti-inflammatory activity. For example, the *V. tropica* venom (5 μg/mL) effectively eliminated lipopolysaccharides-induced activation of NF-κB signaling pathway [136]. Likewise, application of the *Nasonia vitripennis* crude venom (2.5 μg/mL for 8 h) significantly attenuated IL-1β, IL-6, and NF-κB-mediated inflammatory processes [137].

### 3.4. Other Activities

A variety of studies have shown that the venoms of wasps have potent anticoagulant activity [17,138,139]. Several anticoagulant proteins that interact with platelets or coagulation factors have been identified [93,95,140]. However, no anticoagulant bioactive peptide has been identified from that wasp venom. Non-steroidal anti-inflammatory drugs and opioid analgesics suffer drawbacks such as cardiovascular risk and addiction. Some peptides isolated from the wasp venom, such as Agelaia-MPI and Thr^6^-bradykinin evoked potent antinociceptive behavior [39,141,142], suggesting that wasp venom may become an alternative therapy for pain management.

## 4. Wasp Venom Allergy and Immunotherapy

Wasp venoms are complex mixtures containing enzymes, proteins and allergens [143], of which phospholipase A1, hyaluronidase and antigen 5 are the three most important allergens that cause severe allergic reactions [77,100,144]. After being stung by a wasp, the mild reactions may cause local swelling and pain, and the severe ones may generate a systemic allergic response or even endanger life [143,145,146]. Unlike common allergic reactions transmitted through the skin or epithelial cells, the venom of wasps can be injected directly into the blood through the skin, causing rapid systemic transmission [143,147]. The irritant and toxic components of the wasp venom cause pathological phenomena such as pruritus, blisters, swelling, and pain in the sting area of the wasp. More significant local reactions, characterized by a larger area (over 10 cm in diameter) and longer duration (over 24 h, or even 3 to 10 days) as described above, are considered to be some of the major allergic reactions [148,149]. Systemic reactions or allergic reactions include skin urticaria, anaphylactic shock, acute renal failure, central and peripheral nervous system syndrome, bradycardia, arrhythmia, multiple organ dysfunction syndromes, etc. [143,150]. Severe allergic reactions due to wasp stings may lead to rapid death. More than 50 deaths from wasp bites have been recorded annually in the United States, and many more have not been documented [151,152]. Identifying allergens is a critical step in the treatment of venom allergy and is essential for targeted therapeutic interventions [143]. An accurate diagnosis was primarily based on history, skin tests, radioactive allergen sorbent assay (RAST), specific IgE (sIgE) antibody identification, and basophilic activation test (BAT) [153,154,155]. Prescription drugs such as epinephrine, antihistamines and corticosteroids are usually selected to treat an allergy caused by wasp venom. The only available curative treatment is venom allergen immunotherapy (VIT) [143]. While VIT therapy can reduce the risk of systemic reaction to less than 5% with the effectiveness rate up to 95% to 97%,VIT therapy also faces some challenges. VIT includes venom injections with several weeks of metered increases, and a 3–5 years maintenance period, making treatment costly and difficult to maintain [156]. To address these problems, rush or ultrarush treatment protocols have been introduced for up-dosing in just a few hours [156,157,158]. VIT therapy also faces the great challenge of cross-reactivity. The cross-reactivity may be due to the same allergen molecules from different wasp venom sources, or similar antigenic determinants on the wasp venom allergen [58,159,160]. Component-resolved diagnostics using recombinant non-glycosylated allergen components could distinguish the difference between cross-reactivity and true allergy, improving appropriate immunotherapeutic intervention [143]. Recently, the application of omics to venom research has enabled the rapid discovery of new, medically significant venom components to better understand the unique and characteristic components of the entire venom system at the molecular level. With these new technologies, the diagnostic sensitivity of allergic reactions to wasp bites, as well as the accuracy and success rate of patient treatment, will be significantly improved.

## 5. Conclusions

Wasp venoms are intriguing animal venoms due to their therapeutic usefulness. Several bioactive peptides and proteins isolated from wasp venom have been reported regarding their medical application [6,7,8,9,10,11]. Wasp venoms also present challenging therapeutic targets due to the varied chemical contents delivered upon wasp stings. Thus, accumulating functional knowledge on the bioactivity of venomous wasp molecules would improve the use of wasp venoms for medical applications. Additionally, a better understanding of the structure, function, physicochemical properties, and pathology behind these peptide and protein components is critical for developing improved wasp sting remedies and innovative medications.

With the rapid development of genomics, transcriptomics, proteomics, and structural biology, the increasing sensitivity of detection techniques, and our growing understanding of the nature of human health and pathology, our research on animal venom, including that of wasps, has grown faster and more profound. A variety of biological activity in wasp venoms remains unexplored, and some peptides and protein components remain unidentified, uncharacterized, or studied due to the venom’s low concentrations. When wasp venom is thoroughly explored, it will undoubtedly lead to the elucidation of new biological functions and the development of useful research tools, diagnosis, and treatment, ultimately better serving human health.

## 6. Materials and Methods

We sought articles published up to February 2022 from Web of Science, PubMed, and Scopus using the following headings and keywords alone or with various combinations: “wasp venom”, “in vivo studies”, “in vitro studies”, “Therapeutic benefits”, and “Medical application”. Peer-reviewed publications with PubMed unique identifier (PMID), a digital object identifier (DOI), and chapters in books with publishers, publishing times, and standard page numbers, were considered in the current review article.

## Figures and Tables

**Figure 1 biomolecules-12-00527-f001:**
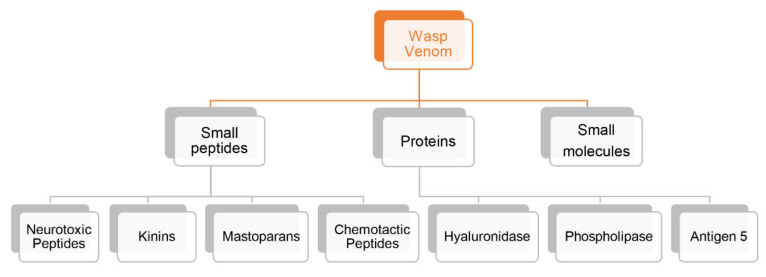
The basic components of wasp venom.

**Figure 2 biomolecules-12-00527-f002:**
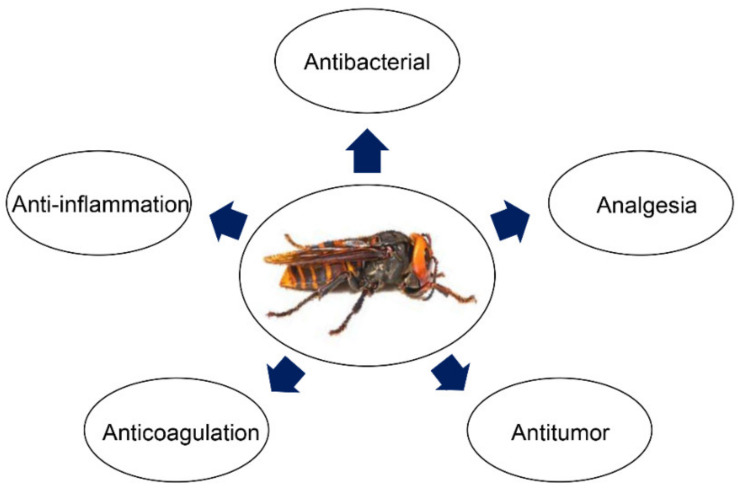
The pharmacological activity of bioactive peptides from wasp venom.

**Table 2 biomolecules-12-00527-t002:** Representative Kinins from WaspVenom.

Wasp-Scientific Name	Isolated Peptides	AA Sequence	References
*Vespa mandarinia*	Vespakinin-M	GRPXGFSPFRID	[42]
*Vespa magnifica*	Vespakinin-M	GRPPGFSPFRID	[41]
*Vespa xanthoptera*	Vespakinin-X	ARPPGFSPFRIV	[43]
*Vespa analis*	Vespakinin-A	GRPPGFSPFRVI	[44]
*Vespa tropica*	Vespakinin-T	GRPPGFSPFRVV	[44]
*Polybia occidentalis*	Thr^6^-bradykinin	RPPGFTPFR	[39]
*Megascolia flavifrons, and Colpa interrupta*	Thr^6^-bradykinin-Lys-Ala	RPPGFTPFRKA	[44,45]
*Cyphononyx fulvognathus and Polybia paulista*	RA-Thr^6^-Bradykinin	RARPPGFTPFR	[46]

**Table 5 biomolecules-12-00527-t005:** Proteins from Wasp-venom.

Protein	Isolated Protein	Wasp-Scientific Name	References
Hyaluronidase	VesA2	*Vespa affinis*	[76]
	Vesp ma 2	*Vespa magnifica*	[77]
	VesT2a	*Vespa tropica*	[78]
	VesT2b	*Vespa tropica*	[76]
	Vesp v 2A	*Vespa velutina*	[76]
	Vesp v 2B	*Vespa velutina*	[76]
Phospholipase	Vesp a 1.1	*Vespa affinis*	[79]
	Vesp a 1.2	*Vespa affinis*	[79]
	Phospholipase A1	*Polybia paulista*	[80,81]
	vPLA2	*Vespa basalis*	[82]
	Phospholipase A1(Ves v 1)	*Polybia paulista*	[80]
	Magnifin (PLA1)	*Vespa* magnifica	[83]
	Orientotoxin I	*Vespa orientalis*	[84]
	Orientotoxin II	*Vespa orientalis*	[84]
	VT 1	*Vespa velutina*	[85]
	Vesp v 1	*Vespa velutina*	[86]
	PLB I	*Vespa xanthoptera*	[87]
	PLB II	*Vespa xanthoptera*	[87]
Antigen 5	Vesp c 5.01	*Vespa crabro*	[88]
	Vesp c 5.02	*Vespa crabro*	[88]
	Vesp m 5	*Vespa mandarinia*	[88]
	Vesp ma 5	*Vespa magnifica*	[89]
	Vesp v 5	*Vespa velutina*	[77]
	Magnvesin	*Vespa magnifica*	[86]
Dipeptidyl Peptidase IV	No name	*Vespa affinis*	[90]
	No name	*Vespa basalis*	[91]
	No name	*Vespa tropica*	[92]
Serine Protease	Bicolin	*Vespa bicolor*	[93]
	Protease I	*Vespa orientalis*	[94]
	No name	*Vespa velutina*	[95]

## Data Availability

No new data were created or analyzed in this study. Data sharing is not applicable to this article.

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
