# Peer review of "Bioactive Peptides and Proteins from Wasp Venoms"

_biomolecules, 2022, doi:10.3390/biom12040527_

Round 1
Reviewer 1 Report
The review “Bioactive peptides and proteins from wasp venoms” summarizes the data about structure and biological activity of wasp venom constituents. This work represents certain interest to toxinologists, however, it has several drawbacks. First of all, the language requires strong revision edition. Some phrases are completely not clear. Other particular comments are listed below.
Page 1, lines 39-42. Please, provide some references.
Lines 42-43. The transition from bees to wasps is unclear.
Page 3, line 86-91. This text should be rewritten. There are no data in the cited works about interaction of kinins with the nicotinic acetylcholine receptor. The kinins block the nicotinic synaptic transmission by depletion of the transmitter at the presynaptic site. They also produce anti-nociceptive effects. E.g., Mortari et al. Inhibition of acute nociceptive responses in rats after i.c.v. injection of Thr6-bradykinin, isolated from the venom of the social wasp, Polybia occidentalis. Br J Pharmacol. 2007 Jul;151(6):860-9.
Line 104. The table should be referenced somewhere in the text.
Lines 113-114. “The bradykinin-like peptide was the first neurotoxic and pain-inducing peptide discovered in wasp venom and found to inhibit nAChR of insects in an irreversible manner.” This phrase is not correct. Please, see the previous comment.
Line 116. [-PPGF (T/S) P(F/L)-] What does this mean?
Page 4, lines 120-121. Please, see the pervious comments.
Line 127. “Since kinins are known to cause pain in vertebrates”. The recent data show that kinins have antinociceptive effect. E.g., Mortari et al., 2007.
Line 131. The table should be referenced somewhere in the text.
Lines 134-135. “Among the peptide components found in hunting wasp venoms, the mastoparans are by far the most prevalent. However, it is only in the venom of Vespidae where these peptides have been identified.” It seems that there is a contradiction between these two phrases.
Lines 139-140. This phrase is not clear.
Lines 141-142. This phrase is not clear too.
Line 145. What is MCD?
Lines 153-154. This phrase is not clear. May “their effect against human erythrocytes” prevent “their cell lytic action against insect cells”?
Page 5, line 156. Mastoparan lysis of what?
Line 165. The table should be referenced somewhere in the text. The table title should be corrected.
Page 6, lines 171-172. In line 136, it is written that “mastoparans are tetradecapeptides”. What is correct? “C-terminal amidated secondary structure” – secondary structure cannot be C-terminal amidated.
Line 186. In single letter amino acid codes, Y is tyrosine. Please, consider using another letter.
Line 201. The table should be referenced somewhere in the text.
Page 7, line 208. The sequence cannot be primary. Please, use either “amino acid sequence” or “primary structure”.
Lines 207-210. Why vespakinin M is considered as other peptide, but not as kinin?
Lines 210-212. Why protonectins are considered as other peptides, but not as chemotactic peptides?
Line 221. Glycoproteins are always glycosylated.
Lines 224-225 “…is known for stimulating the secretion of from hyaluronic acid’s reducing groups.[70]” This phrase is not clear.
Line 235. More potent in what?
Page 8, line 240. Phospholipids should be changed to lysophospholipids.
Lines 240-241. This phrase is not clear. Passage to where? Enema is a very strange symptom.
Line 247. “6–1 percent” This value is not clear.
Line 254. Section 2.2.3. Antigen 5. Some biochemical characteristics of this antigen should be described.
Lines 269-270. What are these genera?
Lines 270-272. This phrase is extremely vague.
Lines 286-288. This phrase is not clear.
Page 9, lines 290-304. English should be corrected in this paragraph.
Line 306. The table should be referenced somewhere in the text.
Line 340. The reference 114 describes AMPs from toad.
Page 10, section 3.1. Antimicrobial Activities. The information about antimicrobial activities is too general. It should be supplemented by some details, including affected bacterial species, acting concentrations, comparison with known antimicrobial agent and so on.
Page 11, section 2.2. Antitumor Activities. Again, very general information is given. Only cytotoxic activity against some cancer cell lines is discussed and no data about antitumor or anticancer activity are presented. Еffective concentrations should be given and compared with those of known drugs.
Page 11, section 3.3. Anti-inflammatory Activities. This section either should be supplemented with the data about anti-inflammatory activity of wasp peptides or proteins or removed from the manuscript. In its present form it does not correspond to the manuscript topic.
Page 13, lines 433-435. Please, provide references describing medical application of bioactive peptides and proteins isolated from wasp venom.
Line 443 “protein omics” is proteomic.
Author Response
Reviewer #1:
Comments and Suggestions for Authors
The review “Bioactive peptides and proteins from wasp venoms” summarizes the data about structure and biological activity of wasp venom constituents. This work represents certain interest to toxinologists, however, it has several drawbacks. First of all, the language requires strong revision edition. Some phrases are completely not clear. Other particular comments are listed below.
Response: We appreciate the reviewer’s comments. In the revised manuscript, we have carefully addressed the concerns of this reviewer, as outlined below.
Page 1, lines 39-42. Please, provide some references.
Lines 42-43. The transition from bees to wasps is unclear.
Response: Thanks for spotting this, we have rephrased the sentence as “The prevention and treatment effects of the wasp venom on rhinitis, rhinoconjunctivitis, rheumatoid arthritis, ischemia stroke, Alzheimer disease, Parkinson disease and epilepsy have been gradually proved”. We have also added more references.
Page 3, line 86-91. This text should be rewritten. There are no data in the cited works about interaction of kinins with the nicotinic acetylcholine receptor. The kinins block the nicotinic synaptic transmission by depletion of the transmitter at the presynaptic site. They also produce anti-nociceptive effects. E.g., Mortari et al. Inhibition of acute nociceptive responses in rats after i.c.v. injection of Thr6-bradykinin, isolated from the venom of the social wasp, Polybia occidentalis. Br J Pharmacol. 2007 Jul;151(6):860-9.
Response: Thanks for spotting it out. We have thoroughly revised the text in line 86-91, which now reads as:“The first neurotoxic component identified from wasp venom was a kinin that irreversibly blocks the nicotinic synaptic transmission by depleting the transmitter at the presynaptic site [23-25]. Progressively, Tom Piek et al. reported a Threonine6-bradykinin in the venom of the wasp Colpa interrupta (F.), which presynaptically blocked nicotinic synaptic trans-mission in the Central Nervous system (CNS) of insect [26]. It has been gradually estab-lished that many kinins contained in Vespidae wasp venom are responsible for paralysis [4, 27]. To date, most of the neurotoxic peptides of hunting wasps are kinins”.
Lines 113-114. “The bradykinin-like peptide was the first neurotoxic and pain-inducing peptide discovered in wasp venom and found to inhibit nAChR of insects in an irreversible manner.” This phrase is not correct. Please, see the previous comment.
Response: The first component discovered in wasp venom was a kinin-like or BK-related peptide (BRP), characterized as a pain-producing molecule [38]. Since then, many peptides from this class have been discovered in social wasp venoms and are generally recognized as wasp kinins.
Page 4, lines 120-121. Please, see the pervious comments.
Response: Thanks for spotting it out. We rephrased the pain-associated description of kinins and added the anti-nociceptive effect of Thr6-bradykinin in line 119-121.
Line 104. The table should be referenced somewhere in the text.
Response: Fixed.
Line 116. [-PPGF (T/S) P(F/L)-] What does this mean?
Response: This is a summary of similar amino acid sequences of bradykinin from wasp and mammals, where P represent proline, G represents glycine, and so on. (T/S) indicates that the site is threonine or serine.
Line 127. “Since kinins are known to cause pain in vertebrates”. The recent data show that kinins have antinociceptive effect. E.g., Mortari et al., 2007.
Response: As a previous response, we rephrased the pain-associated description of kinins.
Line 131. The table should be referenced somewhere in the text.
Response: Fixed.
Lines 134-135. “Among the peptide components found in hunting wasp venoms, the mastoparans are by far the most prevalent. However, it is only in the venom of Vespidae where these peptides have been identified.” It seems that there is a contradiction between these two phrases.
Response: We have rephrased these two phrases as “The mastoparans are the most abundant peptide components of hunting wasp venoms, and are widely distributed in Vespidae: social and solitary.”.
Lines 139-140. This phrase is not clear.
Lines 141-142. This phrase is not clear too.
Response: We have rephrased the text as “Additionally, mastoparans have structural features, including an amphiphilic α-helix structure and a net positive charge, in which all hydrophobic amino acid side chains are on one side and those of basic or hydrophilic amino acid residues on the opposite side. This enables the mastoparans to attach to the biomembranes and form pores, thus increasing cell membrane permeability”.
Line 145. What is MCD?
Response: Fixed. We provided the full name of MCD activity (mast cell degranulating activity).
Lines 153-154. This phrase is not clear. May “their effect against human erythrocytes” prevent “their cell lytic action against insect cells”?
Response: We have rephrased the phrase as “Moreover, mastoparans have been found to cause feeding disorder in lepidopteran larvae, most likely due to their non-specific neurotoxic or myotoxic action induced lytic ac-tion against insect cells”.
Page 5, line 156. Mastoparan lysis of what?
Response: Fixed.
Line 165. The table should be referenced somewhere in the text. The table title should be corrected.
Response: Fixed.
Page 6, lines 171-172. In line 136, it is written that “mastoparans are tetradecapeptides”. What is correct? “C-terminal amidated secondary structure” – secondary structure cannot be C-terminal amidated.
Response: In lines 171-172, we mistakenly wrote chemotactic peptides as mastodons. It has now been corrected. We also modified the description in line 172.
Line 186. In single letter amino acid codes, Y is tyrosine. Please, consider using another letter.
Response: We thank the reviewer for raising this point. We named the hydrophobic amino acid Z and modified the text accordingly.
Line 201. The table should be referenced somewhere in the text.
Response: Fixed.
Page 7, line 208. The sequence cannot be primary. Please, use either “amino acid sequence” or “primary structure”.
Response: Thanks, Fixed.
Lines 207-210. Why vespakinin M is considered as other peptide, but not as kinin?
Response: Thanks for spotting it out. We have modified the text accordingly.
Lines 210-212. Why protonectins are considered as other peptides, but not as chemotactic peptides?
Response: Thanks for spotting it out. We have modified the text accordingly.
Line 221. Glycoproteins are always glycosylated.
Response: Fixed.
Lines 224-225 “…is known for stimulating the secretion of from hyaluronic acid’s reducing groups.[70]” This phrase is not clear.
Response: Fixed.
Line 235. More potent in what?
Response: We mean more potent at inducing allergic reactions. We have elaborated this part accordingly.
Page 8, line 240. Phospholipids should be changed to lysophospholipids.
Response: Fixed.
Lines 240-241. This phrase is not clear. Passage to where? Enema is a very strange symptom.
Response: We have added the following statement to make our intended meaning clear; “Due to the vast array of symptoms mediated by phospholipases, they play an essential role in the insect sting-triggered envenoming process, especially on Hymenoptera venom allergy (HVA). In addition, these molecules have a variety of direct pathophysiological effects, such as necrosis, cell membrane disruption, inflammation, hemolysis, pore formation, platelet aggregation activation (PLA1), and possibly apoptosis (PLA2).”
Line 247. “6–1 percent” This value is not clear.
Response: Thanks for spotting this. The value should be 6-14 percent.
Line 254. Section 2.2.3. Antigen 5. Some biochemical characteristics of this antigen should be described.
Response: Addressed.
Lines 269-270. What are these genera?
Response: These genera are Vespula and Polistes and have been updated in the text, accordingly.
Lines 270-272. This phrase is extremely vague.
Response: Thanks for spotting this. We have revised the sentences to make them meaningful; “Antigen 5 sequence similarity for species within the same genus is 98 percent but lowers to 57 percent when antigen 5s from different genera, for instance, Vespula and Polistes, are compared [85]. Each of them exhibits some degree of immunological cross-reactivity, which is important to define the composition of mixtures for desensitization treatment”.
Lines 286-288. This phrase is not clear.
Response: Addressed.
Page 9, lines 290-304. English should be corrected in this paragraph.
Response: Fixed.
Line 306. The table should be referenced somewhere in the text.
Response: Fixed.
Line 340. The reference 114 describes AMPs from toad.
Response: Thanks. Fixed.
Page 10, section 3.1. Antimicrobial Activities. The information about antimicrobial activities is too general. It should be supplemented by some details, including affected bacterial species, acting concentrations, comparison with known antimicrobial agent and so on.
Response: We appreciate Reviewer’s insight. Detailed information has been supplemented in this section.
Page 11, section 3.2. Antitumor Activities. Again, very general information is given. Only cytotoxic activity against some cancer cell lines is discussed and no data about antitumor or anticancer activity are presented. Еffective concentrations should be given and compared with those of known drugs.
Response: We appreciate Reviewer’s insight. Detailed information has been supplemented in this section.
Page 11, section 3.3. Anti-inflammatory Activities. This section either should be supplemented with the data about anti-inflammatory activity of wasp peptides or proteins or removed from the manuscript. In its present form it does not correspond to the manuscript topic.
Response: Thanks for spotting it out. We have supplemented with the data about anti-inflammatory activity in the text.
Page 13, lines 433-435. Please, provide references describing medical application of bioactive peptides and proteins isolated from wasp venom.
Response: Addressed.
Line 443 “protein omics” is proteomic.
Response: Thanks. Fixed.
Reviewer 2 Report
This paper gives a comprehensive overview of wasp venom peptide and protein components, including potential medical application. This survey should be useful for anybody interested in the field to wasp venom toxins and can direct them to the relevant primary literature. However, some of the language and statements are unclear, and these issues need to be addressed as indicated below before the review is ready for publication. Also, the review would benefit from some more mechanistic/structural detail on toxin activity where available, this would elevate it beyond a simple list of examples and studies.
Minor issues to be addressed:
Line 27: "almost the largest order in insects" - please be more precise
Lines 39-44: Potential uses of bee venom are listed, but the conclusion is made for wasp venom. Also, the different examples are insufficiently supported by the literature quoted. I would suggest to follow more closely the examples given in reference 6 for bee and wasp toxins potential therapeutic applications.
Line 65: remove "and"
Line 80: Is the 50 % an approximate estinate by the authors, or is there a reference for this statement. This should be made clear (and the reference added if available).
Line 91: unclear, are kinins most neurotoxic, or are most neurotoxic peptides kinins?
Lines 96-100: review grammar
Tables 1,2: please do not use justified paragraph format in first column.
Line 127-128: should probably read: "widespread presence in venoms [...] implies" rather than "their widespread venoms [...] imply"
Line 140: amino acid residues on one side the other side, ...
Line 145: spell out MCD at the first mention
Line 152: E. coli should be in italic
Table 3: Why is there a row for Vespula lewisii with no peptide entry? peptide 12d should be capitalized for consistency
Line 185-186: I would use X instead of Y to denote a hydrophobic residue, since Y is also tyrosine.
Line 192: The authors probably mean "similar sequences", not "identical sequences"
Line 207-210: The second sentence is redundant, as that statement is already made in the first sentence.
Line 225: an object seems to be missing between "sectretion of" and "from hyaluronic acid's"
Line 263-264: This sentence is unclear, is the statement that the sequence identity (or similarity) between fire ant and vespid antigen 5 is ~35 %?
Line 295: the end of the sentence should read: "responsible for paralysis against spider"
Line 385: Instead of "limit their use due to", I would suggest writing "suffer drawbacks such as"
Line 422: What are "carbohydrate determinants-free allergens"?
Line 481: reference incomplete
General: some unnecessary capitalizations should be removed (for example, line 83, "chemotactic Peptides" should be "chemotactic peptides")
Author Response
Reviewer #2:
Comments and Suggestions for Authors
This paper gives a comprehensive overview of wasp venom peptide and protein components, including potential medical application. This survey should be useful for anybody interested in the field to wasp venom toxins and can direct them to the relevant primary literature. However, some of the language and statements are unclear, and these issues need to be addressed as indicated below before the review is ready for publication. Also, the review would benefit from some more mechanistic/structural detail on toxin activity where available, this would elevate it beyond a simple list of examples and studies.
Response: We thank the reviewer for the encouraging comments. As discussed in details below, the revision has addressed the concerns raised by the reviewer.
Minor issues to be addressed:
Line 27: "almost the largest order in insects" - please be more precise
Response: Fixed.
Lines 39-44: Potential uses of bee venom are listed, but the conclusion is made for wasp venom. Also, the different examples are insufficiently supported by the literature quoted. I would suggest to follow more closely the examples given in reference 6 for bee and wasp toxins potential therapeutic applications.
Response: Thanks for spotting it out. We have rephrased the text as "The prevention and treatment effects of the wasp venom on rhinitis, rhinoconjunctivitis, rheumatoid arthritis, ischemia stroke, Alzheimer disease, Parkinson disease and epilepsy have been gradually proved [6-11]. Other potential therapeutic activities such as anticancer activities are also being investigated [12]. Therefore, wasp venom is an essential reservoir of pharmacologically active molecules [13]. "
Line 65: remove "and"
Response: Fixed.
Line 80: Is the 50 % an approximate estinate by the authors, or is there a reference for this statement. This should be made clear (and the reference added if available).
Response: Thank you for spotting this. There a reference for this statement we have cited it accordingly.
Line 91: unclear, are kinins most neurotoxic, or are most neurotoxic peptides kinins?
Response: We have rephrased the text as" To date, most of the neurotoxic peptides of hunting wasps are kinins".
Lines 96-100: review grammar
Response: Fixed accordingly.
Tables 1,2: please do not use justified paragraph format in first column.
Response: Fixed.
Line 127-128: should probably read: "widespread presence in venoms [...] implies" rather than "their widespread venoms [...] imply"
Response: Fixed.
Line 140: amino acid residues on one side the other side, ...
Response: Fixed.
Line 145: spell out MCD at the first mention
Response: Fixed.
Line 152: E. coli should be in italic
Response: Fixed.
Table 3: Why is there a row for Vespula lewisii with no peptide entry? peptide 12d should be capitalized for consistency
Response: Mastoparan has been found in both Vespa xanthoptera and Vespula lewisii, and we have changed the format to avoid misinterpretation.
We have capitalized 12d.
Line 185-186: I would use X instead of Y to denote a hydrophobic residue, since Y is also tyrosine.
Response: Thanks. Fixed.
Line 192: The authors probably mean "similar sequences", not "identical sequences"
Response: Thank you for spotting it. Fixed.
Line 207-210: The second sentence is redundant, as that statement is already made in the first sentence.
Response: We have modified the text accordingly.
Line 225: an object seems to be missing between "sectretion of" and "from hyaluronic acid's"
Response: We have rephrased the text as"The vespid hyaluronidase is a member of family 56 of glycosyl hydrolase with en-do-N-acetylhexosaminidase enzymatic specificity [76], and is known for stimulating the secretion of reducing groups from hyaluronic acid [77]. "
Line 263-264: This sentence is unclear, is the statement that the sequence identity (or similarity) between fire ant and vespid antigen 5 is ~35 %?
Response: We have revised the statement to make it clear “Two of the four allergens found in fire ants are homologues to vespid phospholipases and antigen 5. Antigen 5 from fire ant has approximately 35% sequences identify with antigen 5 from vespid.”
Line 295: the end of the sentence should read: "responsible for paralysis against spider"
Response: Thanks. Fixed.
Line 385: Instead of "limit their use due to", I would suggest writing "suffer drawbacks such as"
Response: Thanks. Fixed.
Line 422: What are "carbohydrate determinants-free allergens"?
Response: Cross-reactive carbohydrate determinant structures found in wasp glycoproteins are commonly recognized by IgE antibodies as epitopes that can lead to extensive cross-reactivity and obscure in vitro diagnostic serology results. With the introduction of component-resolved diagnostics, recombinant non-glycosylated components have been utilized to mitigate the risk of extensive cross-reactivity.
We have changed the “carbohydrate determinants-free allergens” to “non-glycosylated allergen components”.
Line 481: reference incomplete
Response: Fixed.
General: some unnecessary capitalizations should be removed (for example, line 83, "chemotactic Peptides" should be "chemotactic peptides")
Response: Thanks. Fixed.
Reviewer 3 Report
The submitted manuscript “Bioactive peptides and proteins from wasp venoms” by Luo et al summarised the advances in bioactive peptides and protein from the venom of wasps and their biological effects. I think this is indeed a good and logical summary. The manuscript is overall well organized and presented and I could add only a few comments (see below).
- Figure 1, compounds can be changed to “small molecules”
- The style of peptide sequences in all the tables is not consistent. Some of them have C-terminal amide, while some do not.
- Page 10 line 313, shall “we show” be “we summarise”
- Page 10, section 3.1, for AMPs introduction, the recent excellent reviews can be included, such as Nat Rev Microbiol (2021). 19, 786–797 DOI 1038/s41579-021-00585-w, Chem. Soc. Rev., 2021,50, 4932-4973 DOI 10.1039/D0CS01026J,)
Author Response
Reviewer #3:
Comments and Suggestions for Authors
The submitted manuscript “Bioactive peptides and proteins from wasp venoms” by Luo et al summarised the advances in bioactive peptides and protein from the venom of wasps and their biological effects. I think this is indeed a good and logical summary. The manuscript is overall well organized and presented and I could add only a few comments (see below).
Response: We thank the reviewer for the highly encouraging comments. As discussed in detail below, we have revised the manuscript to address the concerns raised by the reviewer.
Figure 1, compounds can be changed to “small molecules”
Response: Thanks, Fixed.
The style of peptide sequences in all the tables is not consistent. Some of them have C-terminal amide, while some do not.
Response: Thank you for spotting this. We have unified the format of peptide sequences.
Page 10 line 313, shall “we show” be “we summarise”
Response: Thanks, Fixed.
Page 10, section 3.1, for AMPs introduction, the recent excellent reviews can be included, such as Nat Rev Microbiol (2021). 19, 786–797 DOI 1038/s41579-021-00585-w, Chem. Soc. Rev., 2021,50, 4932-4973 DOI 10.1039/D0CS01026J,)
Response: We agree, and modified the text accordingly.
Round 2
Reviewer 1 Report
Dear authors,
All my comments are addressed. The manuscript is recommended for acceptance.
Reviewer 1 Report
Response: Thanks.